# An Implementation Design of Unified Protocol Architecture for Physical Layer of LoRaWAN End-Nodes

Jean Park [1] and Juyeop Kim [2,*]

1    Department of Computer and Information Science, University of Pensylvania, Philadelphia, PA 19104, USA; hlpark@seas.upenn.edu
2    Department of the Electronics Engineering, Sookmyung Women's University, Seoul 04310, Korea
*    Correspondence: jykim@sookmyung.ac.kr; Tel.: +82-2-710-9005

**Abstract:** LoRa Wide Area Networks (LoRaWAN) can provide a connectivity service to Internet of Things (IoT) for an extremely long run-time and with low power consumption. As the LoRaWAN is extensively applied to various IoT scenarios, LoRaWAN solutions face a flexibility issue in terms of inter-operating with various kinds of LoRa modem hardware and protocol scenarios. In this regard, we design a unified protocol architecture for LoRaWAN physical layer, which can flexibly correspond to various deployment and operational cases. The new protocol architecture includes a hardware abstraction sub-layer, which contains generalized handlers for configuring various kinds of the LoRa modem, and a physical procedure sub-layer that structurally models the physical layer procedures of the LoRaWAN based on Finite State Machine(FSM). We illustrate the flexibility of the new protocol architecture by implementing an extensive feature that enhances the packet reception ratio based on the status of preamble detection. For evaluating the new protocol architecture, we implement the LoRaWAN physical layer protocol on real-time embedded systems and conduct experiments. The experimental results show that the proposed protocol robustly transmits and receives packets and generates little amount of additional burden compared with the conventional open source protocol provided by SemTech.

**Keywords:** LoRa; LoRaWAN; unified architecture; hardware abstraction layer; physical layer; preamble detection

## 1. Introduction

Many products are getting more intelligent by leveraging the recent information and communication technologies. This trend makes us pay attention to Internet of Things (IoT), where the products inter-operate with other products and exchange information through wireless communications. In point of a wireless communications system, a significant issue is to cover use scenarios with respect to the IoT in an efficient way. Apart from data traffic generated by human, the IoT end-nodes tend to generate small-sized packets infrequently in an extremely long time, and the wireless communication system needs additional consideration to deal with this new kind of traffic for the IoT end-nodes. Specifically, the wireless communication system should be able to provide a connectivity service to the IoT end-nodes operating in low activity and to make those perform data transmission and reception efficiently in a power consumption aspect.

For providing connectivity to the IoT end-nodes, many consider a Low-Power Wide Area Networking (LPWAN) whose design is dedicated to the IoT use scenarios. Apart from the conventional wireless communications systems, the LPWAN induces an end-node to consume a little amount of its power, so battery-powered end-nodes can communicate with the network side in a sufficiently long lifetime. A well-known LPWAN is Long Range (LoRa) Wide Area Network (LoRaWAN), which is an open standard developed by LoRa Alliance [1,2]. The LoRaWAN plays the role of an access network and is composed of multiple gateways that relay end-nodes' data to network servers via standard IP connections.

As a means of wireless transmission, the LoRaWAN basically utilizes LoRa modulation proposed by Semtech. Along with the LoRa modulation, the LoRaWAN specifies PHYsical (PHY) layer and Medium Access Control (MAC) layer protocols that are responsible for transferring data on air by configuring and controlling operations of the LoRa modulation. These protocol layers inter-work with the network layer and ultimately enable the end-nodes to connect to remote network servers.

As the previous surveys has examined [3–5], it is remarkable that main access procedures in the LoRaWAN guarantee reliable communication with low power consumption for the end-nodes. By adopting the chirp-based spread spectrum modulation, data receptions of the end-nodes and gateways is robust to narrowband noise and interference in some degree and can be simply implemented [6,7]. The PHY and MAC layer protocols also keep the end-nodes away from monitoring the signal of gateways all the time while they are connected to the network, which makes the end-nodes have long sleep durations. These open-standard layers garner attention as a prospective wireless communication technology, as it can meet the demands on low cost, low battery life and long range.

For recent years, the LoRa and LoRaWAN have been subjects of interest and widely explored by many researchers in an LPWAN perspective. The majority of the research works highlight to analyze the network capacity of the LoRaWAN in a scalable environment where many end-nodes are covered by a gateway. The packet reception ratio and throughput of the LoRaWAN are numerically analyzed [8–10], and the coverage of the LoRaWAN is illustrated in various user-distribution scenarios and parameter configurations [11,12]. The impact of downlink feedback to the performance of the LoRa modulation is analyzed [13], and the analysis of how the LoRaWAN can satisfy the requirements of 5G massive Machine Type Communication (mMTC) is provided [14]. Furthermore, several research works aim to improve the network performance of the LoRaWAN. The network planing framework for improving link performance as well as coverage and the network clustering scheme based on Spreading Factors (SFs) to form a mesh LoRa network are proposed [15,16]. In addition, it is suggested to use the concept of multimodal retransmission timeout to enhance the performance of Transmission Control Protocol (TCP) and optimal transmission policies under the constraint of duty-cycle [17,18].

Many research works also approach the PHY and MAC layer issues in the LoRaWAN. The performance of multiple access in terms of latency and throughput in collision environments were analyzed [19–24]. New data rate configuration schemes have been proposed to enhance packet reception ratio in dense IoT environments [25,26], and time-power multiplexed channels were utilized for achieving capacity enhancement [27]. A grant-free MAC protocol has been proposed to allocate distinct and dedicated slots for odd and even spreading factors [28], and efficient acknowledgement protocols were proposed for enhancing the reliability and scalability [29,30]. A scheduling concept has been introduced to the MAC protocol for improving energy-efficiency and reliability of data transmissions [31,32]. With respect to PHY layer, the numerical results for packet and frame errors of the LoRa modulation were provided in various signal environments [33,34], and the error probability was illustrated when interference exists and the adaptive data rate is utilized [35,36]. Diversity schemes were proposed for enhancing packet error rate [37], and a coherent detection scheme was suggested for enhancing the receiving performance in the presence of interference [38]. In addition, the concept of Non Orthogonal Multiple Access (NOMA) to LoRa modulation is applied to enable multi-user detection [39].

Since the LoRa and LoRaWAN are brought to reasonable maturity in a technical aspect, the recent LoRaWAN research tends to include system implementation for estimating system-level performance or verifying functionality of practical IoT applications. Several research works provide simulation tools for realizing the LoRaWAN in various test scenarios [40–42], and evaluate the scalability of the LoRaWAN systems based on the simulation tools [43–45]. The performance of the LoRaWAN systems applied to industrial, smart city and North America urban scenarios was evaluated [46–48]. In addition, some research works realized the LoRaWAN by implementing it on real-time operating systems.

The coverage and downlink performance have been empirically estimated based on the measurement results obtained from LoRa modules [49–52]. A new replication scheme was proposed for emergency scenarios and its performance was evaluated by implementing it on real-time embedded systems [53]. Semtech SX1272 module and USRP B210 have been utilized for evaluating the link-level performance [54], and the measurement results of packet reception ratio were provided to prove the effect of the proposed transmission parameter selection scheme [55]. In addition, implementation of an agricultural IoT system is presented based on the LoRaWAN and provides the measurement results from the implemented system [56].

As the LoRaWAN is considered for more extensive IoT scenarios, the LoRaWAN systems need to be implemented in more flexible ways for covering various operational scenarios. The LoRaWAN system consequently needs to have a well-structured PHY layer protocol software that can handle all kinds of transmission and reception scenarios. Meanwhile the LoRaWAN specification refers to simple requirements of PHY layer procedures, the PHY layer protocol practically faces many complicated scenarios due to abnormal events at unexpected moments. Therefore, the PHY layer protocol needs to be designed under a well-organized architecture that generalizes various PHY layer procedures and flexibly realizes the complicated scenarios. The conventional research works mainly focus on the LoRaWAN in a theoretical view, and are lacking in considering robust operations of the LoRaWAN protocol in an implementation perspective. Furthermore, the conventional open source code for the LoRaWAN protocols, including the LoRaMAC-in-C(LMIC) library, does not contain such a unified architecture that can adapt to various scenarios [57].

The PHY layer protocol also has a significant role of configuring and managing operations of the LoRa modulation hardware. According to the regional requirement of the LoRaWAN end-nodes, there are various hardware solutions of the LoRa modulation which support different frequency bands. Furthermore, the LoRaWAN specification is updated with including new features or implementation algorithms, so the hardware solutions of the LoRa modulation can be further revised and newly launched. Therefore, the PHY layer protocol needs to inter-operate with various kinds of the hardware solutions and to have a sufficient level of flexibility for adapting to those solutions efficiently. The conventional PHY protocol dedicated to a specific hardware solution is not desirable in the management perspective since a different set of source code is required for generating software binary for the specific hardware solution. It will be a great burden if we are required to manage a dedicated set of source code for every hardware release or change, which frequently happens during commercialization or commercial operations of IoT systems.

Encompassing the above issues, we aim to design a unified protocol architecture that can flexibly operate in various scenarios of the LoRaWAN PHY layer. We locate sub-layers in the PHY layer protocol and address how the sub-layers can efficiently realize PHY layer procedures and manage the hardware solutions of the LoRa modulation. The major parts of this paper lie in designing LoRa Hardware Abstraction sub-Layer (LHAL) for interfacing with hardware solutions and LoRa PHYsical sub-Layer (LPHY) for covering PHY layer procedures according to requests from the MAC layer. We also present test experiments implementing the proposed PHY layer protocol and show that it performs stably in the aspect of packet reception ratio and delay time. In addition, we implement an additional standard feature to the proposed PHY layer protocol to show that the new feature can be easily implemented to the protocol architecture. The source code implemented for the proposed LoRaWAN PHY layer can be accessed in the open source platform [58].

## 2. System Model and Requirements of the PHY Layer Protocol

Figure 1 illustrates the overall system of a LoRaWAN end-node that we assume as the target of the protocol design. A LoRa modemis an embedded system that has a dedicated role of generating and processing LoRa-modulated signals according to external commands of transmission (TX) and reception (RX). The protocol stack controls the whole end-node operations to comply with the LoRaWAN specification [2], and operates as software on a

platform system independent with the LoRa modem. PHY layer protocol, located within the protocol stack, serves as a bridge between the upper layer protocol and the LoRa modem. At the requests from the upper layer, the PHY layer protocol directly controls the LoRa modem to transmit or receive in the way complying with the LoRaWAN PHY layer procedures. The PHY layer protocol can command operations (i.e., transmission and reception) to the LoRa modem through a hardware interface, such as Serial-to-Peripheral Interface (SPI), provided by the platform system of the protocol stack.

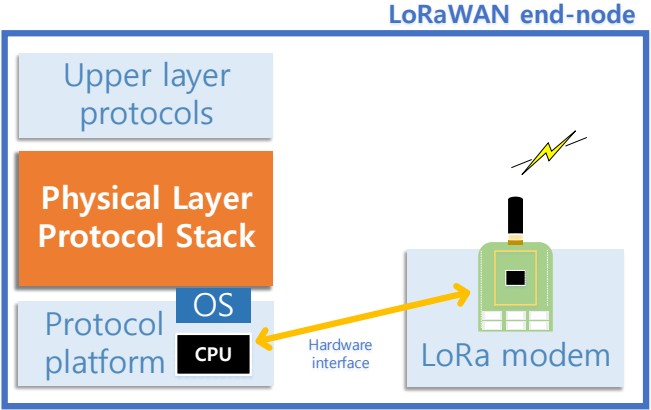

**Figure 1.** The system architecture of the LoRaWAN end-node.

There are various hardware models which can play a role of the LoRa modem and the end-node can be composed of any hardware model. In this circumstance, we assume that the protocol stack should consistently support the functionality of whichever hardware model is attached as the LoRa modem. In this paper, we choose two hardware models, SX1272 and SX1276, as the targets that the protocol stack supports, since they support the unlicensed frequency band for IoT usages in South Korea. Although the protocol stack is designed for only two models here, our design aims to have a framework that can simply extend to support other hardware models of the LoRa modem.

The ultimate goal of our design is to have a PHY layer protocol that consistently works with any request scenarios from the upper layer and hardware model of the LoRa modem. The PHY layer protocol should conduct stable operations that comply with the LoRaWAN specification regardless of the attached LoRa modem hardware or request scenarios. In an upper layer aspect, the PHY layer protocol should start its operations from generalized service functions that characterize the PHY layer procedures. The service functions basically make the PHY layer protocol to trigger TX and RX, flexibly configuring the physical parameters of the LoRa modem hardware. The PHY layer protocol is also required to cope with the multiple service requests in a multi-tasking sense. In addition, the PHY layer protocol needs to have error-handling mechanisms for protecting the LoRa modem hardware from erroneous operations.

In a LoRa modem aspect, the PHY layer protocol needs to adapt to various hardware models whose inner mechanisms and interfaces differ with each other. This means that a unified source code can generate multiple kinds of binaries based on different build options for target hardware models, and each binary can control the corresponding hardware model of the LoRa modem. In detail, the PHY layer protocol should be able to control TX and RX operations of the LoRa modem by issuing commands suitable to the target hardware model. This requires the PHY layer protocol to have a well-defined software architecture that can realize co-existence of hardware-common and hardware-specific functions. Furthermore, the software architecture of the PHY layer protocol is required to be minimally changed when supporting additional hardware models or new protocol features that require inter-operation with the LoRa modem.

*The Architecture of the PHY Layer Protocol*

Based on the above requirements, we consider the PHY layer protocol to have two sub-layers as shown in Figure 2. The functionality of the PHY layer protocol is further divided into the two sub-layers. The LPHY, the upper sub-layer in the PHY layer protocol, manages the status and operations of the LoRa modem at an abstract level. The LPHY includes the implementation of the PHY layer procedures and is independent of the attached LoRa modem. The LHAL is the lower sub-layer of the LPHY and has hardware-specific functions for controlling the LoRa modem. The LHAL directly accesses the hardware resource of the attached LoRa modem according to service primitives given from the LPHY. (We further called the service primitives *LHAL commands*.) By the LHAL commands, the LHAL configures and triggers TX and RX operations of the LoRa modem through the hardware interface.

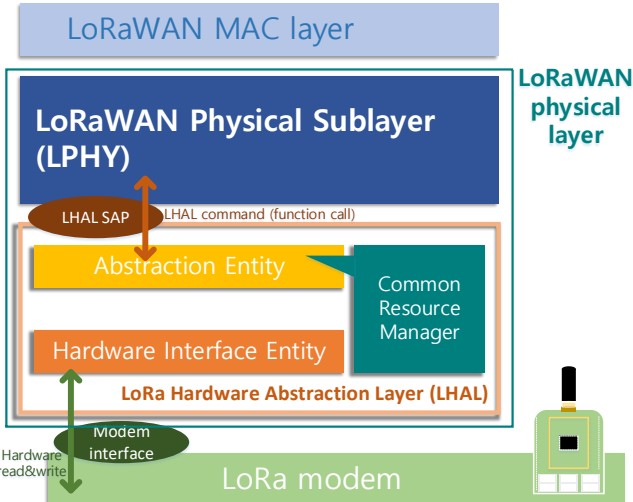

**Figure 2.** The layered architecture of the PHY layer protocol.

Under this sub-layer architecture, hardware-dependent implementation part is separated from the LPHY and this is advantageous in various aspects. The implementation of the PHY layer procedure in the LPHY can be commonly applied to various hardware models of the LoRa modem. Under this architecture, the PHY layer protocol easily adapts to new hardware models by modifying the implementation of the LHAL only. In addition, the functional division by the sub-layers helps us implement more robust protocols, since each sub-layer with a dedicated role will experience simplified service scenarios and can further be implemented with a simple structure.

## 3. LoRaWAN Hardware Abstraction Sub-Layer

Since the LHAL should manage various hardware models of the LoRa modem, we need to consider how the hardware resources of the LoRa modem can be abstracted in a general form prior to LHAL design. The LoRa modem generally has TX-specific and RX-specific hardware, and also some hardware commonly used for both TX and RX, such as Radio Frequency (RF) signal processing. This consequently inspire us to separate the functional parts of the LHAL into TX, RX and common domains. The LHAL functions in the TX and RX domains take care of the TX-specific and RX-specific hardware, respectively. The common hardware needs to be dealt in contention by TX and RX procedures, so the LHAL functions in the common domain manage the common hardware by arbitrating in the accesses to the common hardware by the TX and RX procedures. The function group in the common domain is therefore denoted by a *common resource manager* in Figure 2.

As illustrated in Figure 2, the overall LHAL is vertically divided into abstraction and hardware-interface entities. The abstraction entity manages the *abstracted* factors of the LoRa modem, such as status and configuration parameters, and directly handles LHAL

commands from the upper sub-layer. It commonly checks the validity of incoming LHAL commands based on the current status of the LoRa modem for keeping the LoRa modem away from erroneous operations due to invalid control. On the other hand, the hardware-interface entity is responsible for direct control of the LoRa modem hardware. It provides hardware-specific functions of configuring and operating the LoRa modem hardware to the abstraction entity. Once the LHAL receives an LHAL command from the upper sub-layer, the abstraction entity first analyzes the LHAL command in the hardware-independent aspect. Then, the abstraction entity calls proper hardware-specific functions provided by the hardware-interface entity for triggering proper operation of the LoRa modem hardware.

Table 1 shows the full set of the LHAL commands. The LoRaWAN procedures require the PHY layer protocol to configure the TX and RX parameters adaptively. Our design of the LHAL commands enables to differentiate the parameters such as center frequency, bandwidth, spreading factor code rate and TX power for each TX or RX procedure. Before LHALcmd_transmit or LHALcmd_receive command, LHALcmd_setTxConfig or LHALcmd_setRxConfig can be invoked for operating TX or RX with a specific parameter configurations. These LHAL command definitions are independent with the hardware model of the LoRa modem, since it can be commonly applied to the LoRaWAN procedures with whichever hardware model used as the LoRa modem.

**Table 1.** The LHAL command list.

| Command Name | Description | Parameters |
| --- | --- | --- |
| LHALcmd_transmit | Start TX | Data, size |
| LHALcmd_receive | Start RX | - |
| LHALcmd_setTxConfig | Configure TX parameters | Center frequency, bandwidth, spreading factor, code rate, power |
| LHALcmd_setRxConfig | Configure RX parameters | Center frequency, bandwidth, spreading factor, code rate |
| LHALcmd_query | Query the current state of the LoRa modem | - |
| LHALcmd_abort | Stop the current operation of the LoRa modem | - |

*3.1. Abstraction Entity*

Figure 3 illustrates the detailed mechanism of the LHAL and interaction between the LHAL entities. The abstraction entity operates based on the interactions among the TX, RX and common objects. Incoming LHAL commands let the TX and RX objects to configure TX and RX parameters, respectively, or to make the LoRa modem start its operations. Since the LoRa modem can only do TX or RX at a moment, each LoRa modem hardware has a common register set for the RF signal processing and this register set is configured in both TX and RX cases. So the TX and RX objects access to the common register set, and the common object arbitrates in using the common register set between the TX and RX objects. The common object can only access to the functions of the hardware-interface entity in the abstraction entity, and the TX and RX objects access to the hardware resource through the common object.

To access to the hardware resource, the TX or RX object requests to the common object *to reserve* the hardware resource. The common object allows the reservation of the hardware resource if it is not currently occupied and then the TX or RX object can configure or trigger operations of the LoRa modem. This reservation mechanism keeps the LoRa modem away

from duplicated configurations and operations by the TX and RX objects. (The common object exceptionally allows to override TX operation during RX.) This is essential when the upper layer is multi-threaded and TX and RX LHAL commands may simultaneously arrive at a moment.

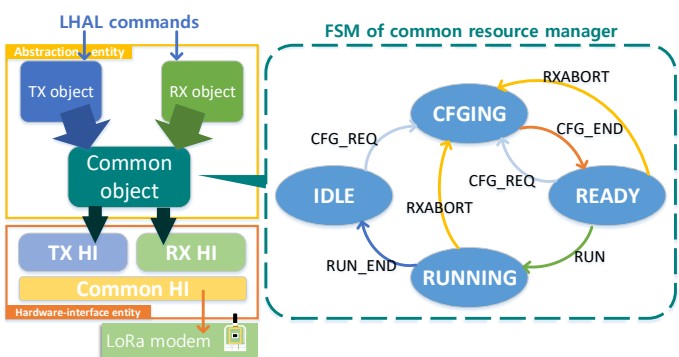

**Figure 3.** The inner mechanism of the LHAL.

Since the arbitration is driven by events of TX and RX procedures, we designed the common object based on Finite State Machine (FSM). The right hand side of Figure 3 shows the detailed mechanism of the common object expressed by state and transitions of the FSM. Each state in the FSM represents the operation and configuration status of the LoRa modem. The common object is initially in *IDLE* state, in which the LoRa modem is not performing any operation. When the TX or RX object requests to configure parameters, the common object reserves the hardware resource and transits to *CFGING* state. In the *CFGING* state, the common object does not further allow other reservations of the hardware resource. After the hardware resource is completely configured, the common object goes to *READY* state, in which the common object allows *RUN* event and start of TX or RX operation. When the TX or RX object triggers operation, the common object controls the LoRa modem to make the operation and goes to *RUNNING* state. The common object in the *READY* state also allows additional configurations, which causes it to go back to the *CFGING* state. After the LoRa modem finishes TX or RX operation, the common object returns to the *IDLE* state.

For protecting the LoRa modem from erroneous controls, the common object only handles proper events based on its state. For example, the common object in the *CFGING* state ignores *CFG_REQ* event so that it prevents duplicated configurations by rejecting the latter one. The common object in the *IDLE* state does not accept the *RUN* event so that it prevents the LoRa modem to start operation without configuration. In addition, the upper layer may want to stop the current RX operation when it is occasionally required to send a data packet. The common object in the *RUNNING* state can transit to the *CFGING* state as it gets a proper LHAL command and makes the LoRa modem abort the previous RX operation. The common object then goes to the *READY* state once the LoRa modem has finished the abortion and is ready for the next operation.

This abstraction entity characterizes the proposed protocol architecture and differentiates from the HAL in the conventional open source. In case of the LMIC library, the HAL is implemented in a sequential structure, and the code for hardware control is sequentially executed. This HAL code only allows us to process a single request, and operates inefficiently when the upper layer gives multiple requests. For example, the LMIC program stays at the middle of a HAL code when the HAL commands and waits for response of the LoRa modem. In this status, the upper layer cannot command further action to the HAL, which is a constraint on implementing more complicated PHY procedures such as to abort RX and to trigger the next TX. On the other hands, the HAL operations under the proposed protocol architecture are proceeded based on the event-driven abstraction entity, and can process upper layer's requests in more complicated scenarios.

### 3.2. Hardware-Interface Entity

The hardware-interface entity contains hardware-specific functions for controlling the LoRa modem hardware. It serves the common object by providing the methods of configuring and triggering operations of the LoRa modem hardware and triggering operations. Since the abstraction entity needs to be independently implemented with hardware models of the LoRa modem, the prototypes of the hardware-specific functions are essentially in a consistent form and regardless of the LoRa modem hardware. The function bodies defined in the hardware-interface entity only contains hardware-dependent implementation so that it can properly control the LoRa modem according to the hardware model.

To design the set of function prototypes, we analyze the register maps of the target hardware models and categorize it in the perspective of PHY layer procedures. Table 2 shows the categories for the register sets of SX1272 and SX1276. Among the whole register maps [59,60], we have chosen the essential ones for realizing the LoRaWAN PHY layer procedures and grouped them as a category if they are potentially dealt during a specific procedure. Most of the registers except for those in the category *freqParms* and *encParms* commonly exist in SX1272 and SX1276. Therefore, the handler functions for the most of the registers can be commonly implemented for both SX1272 and SX1276. The other handler functions for the hardware-specific registers in *freqParms* and *encParms* should be dedicated to each hardware model and separately implemented.

**Table 2.** Categorization of the registers of SX1272 and SX1276.

| Category | Description | Related SX1272/SX1276 Registers |
| --- | --- | --- |
| opMode | operation command | RegOpMode[2:0] |
| freqParms | carrier frequency bandwidth | RegFrMsb, RegFrMib, RegFrLsb RegModemConfig1[7:6] (SX1272), RegModemConfig1[7:4] (SX1276) |
| modParms | modulation options  spreading factor | RegHopPeriod RegPreambleMsb RegPreambleLsb RegModemConfig2[7:4] |
| encParms | decoder parameters (code rate, CRC, and so on) | RegModemConfig1[5:1] (SX1272), RegModemConfig1[3:0], RegModemConfig2[2] (SX1276) |
| txLen | TX payload length | RegPayloadLength |
| txPow | TX power | RegPaConfig |
| txBuf | TX buffer address | RegFifoTxBaseAddr |
| rxBuf | RX buffer address | RegFifoRxBaseAddr |

From the above observation, we designed the function prototypes of the hardware-interface entity as in Table 3. Each function configures registers in a category and provides the abstraction entity a minimal way to control the LoRa modem. The abstraction entity initially configures TX or RX parameters by calling HW_SetTxConfig or HW_SetRxConfig and then triggers operation by calling HW_Tx or HW_Rx. When the upper layer needs to stop the on-going RX for starting new TX, the abstraction entity calls HW_sleep to make sure that the LoRa modem finishes up the previous RX operation, and consequently calls HW_SetTXConfig and HW_TX. In addition, the function HW_setRfTxPower is provided for changing TX power adaptively. These hardware-specific functions enable the abstraction entity to control the LoRa modem without the knowledge of complicated register maps that depend on the hardware model of the LoRa modem. Compared with the HAL in the LMIC library, the hardware-interface entity provides various function prototypes that enable more detailed control of the LoRa modem.

**Table 3.** The function prototypes of the hardware-interface entity.

| API Functions | Description | Related Registers |
|---|---|---|
| HW_Sleep | Force to stop the current operation | opMode |
| HW_init_radio | initialize the hardware | all |
| HW_Rx | start RX | opMode, rxBuf |
| HW_SetRxConfig | configure RX parameters | freqParms, modParms, enParms, rxBuf |
| HW_GetRxPayload | get RX payload | void |
| HW_Tx | start TX | opMode |
| HW_SetTxConfig | configure TX parameters | freqParms, modParms, enParms |
| HW_SetTxPayload | put TX payload | txBuf, txLen, |
| HW_SetRfTxPower | configure TX power | txPow |

For efficient code implementation for the hardware-dependent parts, we utilized a macro as a build option for each hardware model. The build option enables us to select a suitable hardware-dependent code at build time and to obtain a compatible binary to the target hardware model. Figure 4 shows the example of the hardware-dependent code in the HW_setTxConfig function. The code for handling the registers common to both SX1272 and SX1276 is enclosed with *BVARIANT_SX1272* and *BVARIANT_SX1276* macros. Some parts of the code dealing *encParms* and *freqParms* registers need to be hardware-dependent and are separately implemented according to target hardware models. In the case of supporting a new hardware model, we can define a new macro as the build option of the new hardware model and simply add the additional handler code dedicated to the new hardware model.

**Figure 4.** The implementation of the HW_setTxConfig function.

## 4. LPHY: LoRaWAN Physical Procedure Sublayer

The LPHY implementation needs to realize the PHY layer procedures which differentiate with class A, B, and C. Basic procedures follow the class A, which aims for low-power operations and is essentially implemented to every end-node. The class A allows bi-directional communication between an end-node and a gateway when the end-

node *opens* one of the two downlink RX Windows (RXWs) after sending an uplink packet. Before opening each RXW, the end-node needs to have a Receive Delay (RD), and there are two distinct RDs for starting the two downlink RXWs at different moments. (These RDs are further denoted by RD1 and RD2.)

According to the LoRaWAN regional specification, the time difference between RD1 and RD2 should be 1 s. Whereas downlink configuration in the first RXW should be the same as the one in the previous uplink transmission, the gateway can independently configure the center frequency and data rate in the second RXW. (The configuration in the second RXW is robustly set in general.) The two RXWs should last for a certain moment until the LoRa modem effectively detects downlink preamble. After sending an uplink packet, the end-node can transmit the next uplink packet only when it receives a downlink packet in any of the two RXWs or the second RXW ends.

The class B and C are optional implementation to end-nodes and are compatible with class A. The class B end-node is optimized to receive downlink packets addressed by network servers as it is capable of opening RXWs periodically. The class C end-node consumes the majority of time in receiving packets since it continuously listens to downlink channel with the configuration of the second RXW until the end-node sends the next uplink packet. The class B and C procedures can be seen as the variants of the class A RX procedure, so the additional classes can be realized by modifying the RX procedure of the class A.

Figure 5 illustrates the layered architecture and Service Access Points (SAPs) for realizing the class A, B and C procedures. The LPHY plays a role of handling various TX and RX requests from the upper layer and should robustly operate in any request scenarios triggered by the upper layer. The operations of the LPHY are driven by the events from the three sources; service primitives from the upper layer, service responses from the LHAL and internal events triggered by itself. According to the needs of TX or RX, the upper layer can invoke the events through the LPHY SAP by providing service primitives to the LPHY. The LHAL can also invoke the events as responses of the LHAL commands previously invoked by the LPHY. In addition, the LPHY can trigger the internal events for managing on-going operations without any events invoked by the upper or lower layer.

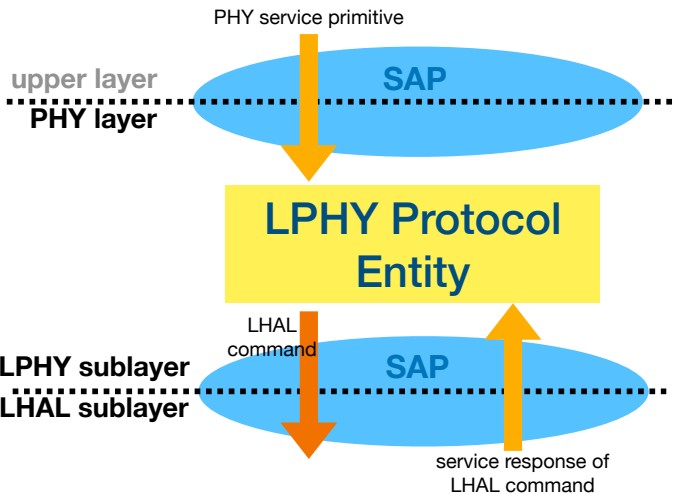

**Figure 5.** The service access points nearby the LPHY.

### 4.1. Designing LPHY Finite State Machine

We start to design the basic LPHY FSM, which realizes the class A procedures and then extends the FSM to cover the class B and C procedures. We propose the basic LPHY FSM that supports the complete set of the class A procedures as shown in Figure 6. For simple implementation, the LPHY FSM consists of the minimal number of the states: IDLE, TX RUN, RX WAIT, and RX RUN. In the IDLE state, the PHY layer is not in TX nor RX procedure and the LoRa modem is not doing any operation. In the TX RUN and RX RUN

states, the PHY layer is in either TX or RX procedure and the LPHY has issued an LHAL command to make the LoRa modem operate. In the RX WAIT state, the LoRa modem is not doing any operation, but the LPHY is scheduled to do RX at a certain moment.

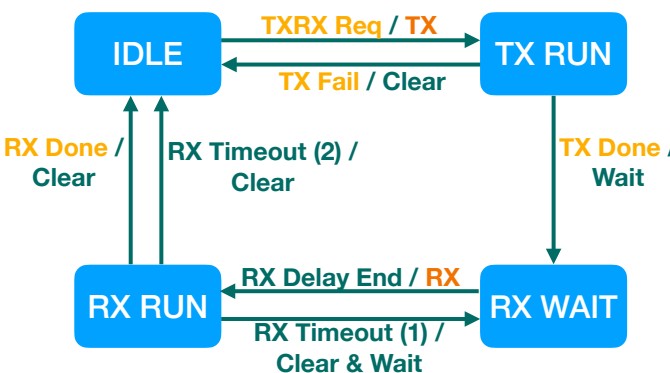

**Figure 6.** The LPHY FSM for the class A operations.

We additionally consider events in the class A scenarios for defining the transitions among the states. In case of the class A, an RX procedure is always followed by a TX procedure, so the LPHY only need to handle one service primitive that requests TX and following two RXs. This event is denoted by *TXRX Req*. The LPHY essentially starts an RX operation RD1 or RD2 after the TX operation, so the LPHY FSM needs to have an internal event for starting RX operations, which is denoted by *RX Delay End*. The LPHY FSM also experiences the events from the LoRa modem, which appear as service responses from the LHAL. In detail, the LPHY FSM can have four LHAL events; *TX Done*, *TX Fail*, *RX Done* and *RX Timeout*. The TX Done event happens after the LoRa modem successfully transmits a packet and the TX Fail happens if the LoRa modem cannot send a packet and aborts the TX operation. In addition, the LHAL triggers the RX Done event when the LoRa modem has successfully decoded a packet, meanwhile it triggers the RX Timeout event when the LoRa modem has not decoded any packet for a given interval.

Based on the states and events, the FSM design can be finished up by defining transitions for every state and event occurrence. The LPHY begins from the IDLE state and transits to the other state when the TXRX Req event happens. The service primitive makes the LPHY initiate TX and opens two consequent RXWs after the TX Done event, so this event triggers a transition from the IDLE state to the TX RUN state. When the TX Done event happens in the TX RUN state, the state is changed to the RX WAIT state for waiting until an RXW opens. Unlike in the IDLE state, the LPHY in the RX WAIT state ignores the pending TX operation for receiving the upcoming downlink packet. When a timer expires and the RX Delay End event happens, the state is changed to the RX RUN with starting RX. If a downlink packet is received, then the state returns to the IDLE for not opening the second RXW. If no packet is received in the first RXW, the LPHY needs to look for the second RXW and the state is changed to the RX WAIT and return to the RX RUN for opening the second RXW. The second RX timeout event lets the state change back to the IDLE.

For each transition, the LPHY needs to perform a proper action. When entering to the TX RUN and RX RUN states, the LPHY triggers *TX* and *RX* operations by issuing the LHAL commands. *Wait* action starts a timer at the entry of the RX WAIT state, and *Clear* action initializes the status of the LoRa modem for preparing the next operation at the entry of the IDLE state. For generalizing RX procedures in the RX WAIT state, we use state variables *RXW count* and *RD timer length*, which memorize the remained number of RXWs and the time interval that the LPHY needs to wait for the next RXW. In the class A scenarios, the RXW count is initially configured as 2 at the moment of the TXRX Req event, and is decreased by one at the end of an RX operation. At each exit of the RX RUN state,

the state goes to the RX WAIT state if the RXW count is positive, and returns back to the IDLE state if the RXW count is 0.

To support the class B and C scenarios, the LPHY should be able to handle other service primitives for realizing different TX and RX patterns. Unlike the class A, the class B additionally requires doing RX at the moments scheduled by the upper layer. This causes it to add a new service primitive to initiate RX only and this event is denoted by *RX Req*. For simple implementation, we consider that the LPHY basically handles this event in the IDLE state. In addition, we can also consider the LPHY to handle this event in RX WAIT or RX RUN for enhancing the RX performance. In the RX WAIT or RX RUN state, this event triggers *Adjust RXW* action that adjusts the current RXW configuration, such as window interval and offset.

To support the class C, the LPHY should be able to do RX continuously after the TX operation, and this is simply realized by extending the action after the TXRX Req event. During an RX procedure, the LPHY further manages and checks a new state variable *RX type*, which indicates the class type with respect to the previous TXRX Req event. When the LPHY recognizes from the service primitive that the TXRX Req event directs for class C, it configures the RXW count as one, the length of the RD timer as 0 and the RX type as class C. This configuration of the state variables makes the state of the LPHY immediately transit from the RX WAIT state to the RX RUN state for immediate and continuous RX. The LPHY FSM also needs to have additional transitions for handling events in the RX procedure. When the TXRX Req event happens during the continuous RX, the LPHY FSM transits from the RX RUN state to the IDLE state for aborting the on-going RX and starts a new TX procedure. In addition, when a downlink packet is received and the RX Done event occurs in the RX RUN state, the LPHY still needs to keep the RX procedure and the the LPHY FSM returns its state to the RX RUN state again.

As a result, the overall LPHY FSM is organized as Table 4. The remarkable point of this FSM design is that the complete scenarios of the classes are realized with the four states. Furthermore, we can see that the LPHY can support the class B and C by minimally adding the events and transitions to the basic FSM for the class A. In terms of securing robustness, this FSM design inspires us to consider the details about unreachable actions and deadlock and help to prevent unexpected termination of the LPHY before an implementation stage.

Furthermore, this FSM design has a sufficient level of flexibility for extending itself to have new features and to realize new PHY layer procedures. Compared with the LMIC library where the PHY layer procedures are realized in a time-sequential way, this FSM design realizes the PHY layer procedures in an event-driven way. This enables the LPHY to be easily modified for adding a new PHY layer procedure or changing the existing PHY layer procedures. We will show how this FSM design is flexible in detail by providing an example of implementing a new standard feature in the next section.

*4.2. Code Implementation*

Based on the LPHY FSM design, we have implemented LPHY sub-layer code. The LPHY FSM operates based on function call due to a service primitive from the upper layer or a service response from the LHAL. Furthermore, the LPHY FSM operates based on an internal event triggered by software timer expiry. The software timer starts at the moment of RX WAIT state entry for realizing RD, and is configured to expire after 1 s and 2 s for the first and second RXW. (This can be differently configured by the upper layer based on a message from a gateway.)

We also utilize other software timers for handling unwanted situations and keeping the LPHY from malfunctioning. These watchdog timers prevent deadlock and trigger forced shutdown of the ongoing procedure of the LoRa modem. In detail, there are two watchdog software timers; TX and RX timers, for monitoring the activity of TX and RX operations. TX timer starts at every entry of the TX RUN state and normally stops when the TX Done or TX Fail event occurs. If the TX timer expires, the LPHY regards it as the TX Fail event and makes the LoRa modem stop the TX operation by performing the Clear

action. This keeps the LPHY from stalling due to the unexpected stall of the LoRa modem during TX operation. The RX timer starts at every entry of the RX RUN state and normally stops at the RX Done event. Expiry of the RX timer means that there is no detected packet and the LPHY regard this as the RX Timeout event.

**Table 4.** State transitions in the LPHY FSM for class A, B and C operations.

| Class | Transition | Event | Condition | Action |
|---|---|---|---|---|
| A | IDLE → TX RUN | TXRX Req | | TX |
| | TX RUN → IDLE | TX Fail | | Clear |
| | TX RUN → RX WAIT | TX Done | | Wait |
| | RX WAIT → RX RUN | RX Delay End | | RX |
| | RX RUN → RX WAIT | RX Timeout | RDW count > 0 | Wait |
| | RX RUN → IDLE | RX Done | RX type = class A | Clear |
| | RX RUN → IDLE | RX Delay End | RDW count = 0 | Clear |
| B | IDLE → RX WAIT | RX Req | | Wait |
| | RX WAIT → RX WAIT | RX Req | | Adjust RXW |
| | RX RUN → RX RUN | RX Req | | Adjust RXW |
| C | RX RUN → RX RUN | RX Done | RX type = classC | RX |
| | RX RUN → IDLE | TXRX Req | RX type = classC | Clear |

The other implementation issue is to take care of simultaneous events, and we utilize event bit flags for handling those events. The LPHY FSM encountering multiple event occurrences needs to handle the events with proper sequence based on its state. The LPHY FSM therefore needs to memorize all the events that have been occurred and are not handled yet, and it needs to have a state variable of event bit flags, each of which memorizes an event occurrence. The bit positions of the event bit flags and corresponding events are listed in Table 5. An event bit is set to 1 when the corresponding event happens, and is cleared to 0 after the event is handled by the LPHY FSM. At the moment of state transition, the LPHY FSM can recognize which events have just occurred by checking each event bit flag.

**Table 5.** Event bit flags.

| Bit Pos | Event Name | Description |
|---|---|---|
| 7 | RX Timeout | End of the RXW. |
| 6 | RX Done | A packet reception by LoRa modem. |
| 5 | RX Delay End | End to wait for the next RXW. |
| 4 | RX Req | Class B RX request from upper layer |
| 3 | TX Done | Transmission failure by LoRa modem. |
| 2 | TX Fail | Successful transmission by LoRa modem. |
| 1 | Reserved | - |
| 0 | TXRX Req | Class A/C TX and RX requests from upper layer. |

The event bit flags make it easy to identify scenarios of multiple event occurrences and to handle each of the events in prior. For example, the LoRa modem may receive a downlink packet and the RX Done event is indicated from the LHAL, and at the same time, the RXW ends and the RX timeout event happens. In this case, the LPHY FSM can recognize that both of the events occur by checking the event bit flags, and the LPHY can process the RX Done event in prior so that it can ensure the packet reception. On the other hands, if the LPHY FSM only memorizes a single event and the RX Timeout event occurs latter, the LPHY misunderstand that no packet is received and the received packet can be unnecessarily discarded.

## 5. Implementing a RXW Prolonging Scheme Based on Preamble Detection

To validate the flexibility of the proposed protocol architecture, we implement a standard feature which can enhance the overall RX performance. An RX issue comes up when transmission time on air gets longer according to parameter configuration. With respect to the TX configuration parameters, the transmission time $T$ is induced as following,

$$T = \frac{S}{SF \times \frac{BW}{2^{SF}} \times CR} sec, \tag{1}$$

where $S$ is the packet size, $SF$ is the spreading factor, $BW$ is the bandwidth, and $CR$ is the coding rate. Meanwhile $S$ is given from the upper layer and channel coding scheme and $BW$ are fixed in practical, $SF$ can vary in time and effectively impact the transmission time.

With our protocol design, the end-node suffers a problem during packet reception if the transmission time is similar with or longer than RXW size. When $SF$ is configured with a large value, the transmission time becomes longer than 1 s, which is the default size of the first RXW. In this case, even the gateway starts sending a downlink packet at the beginning of the first RXW, it continues to transmit a downlink packet after the first RXW ends. According to the RX procedure in the LPHY FSM, the RX timeout event happens and the LPHY lets the LoRa modem stop the RX operation while it is receiving the downlink packet. The LoRa modem then aborts the RX operation and the LPHY ends up with missing the downlink packet. This issue also happens when the gateway uses small $SF$ and sends a downlink packet after a certain delay so that the transmission time interval exceeds the ending boundary of the RXW.

The gateway may take a possible solution by configuring long RXW when transmitting with large $SF$ so that the transmission time interval is included in the first RXW. However, this solution is not feasible since the RXW size is configured during connection establishment of the upper layer or with a default value and cannot be adaptively configured for each downlink transmission. The other way is to utilize long RXW as default, but this requires the end-node which should receive downlink packets in the second RXW to allocate large amount of time and memory resources. A better solution is to make the LPHY sub-layer consider preamble detection phase of the LoRa modem in a cross-layer perspective. According to the LoRaWAN specification, the end-node should prolong the RXW if it detects a preamble before the RXW ends. In case the RX Timeout event happens at the middle of the packet reception, the LPHY FSM is conventionally unaware of the downlink packet arrival and stops the RX operation. However, if the LPHY waits for an extra time, the LoRa modem can finish up the packet reception and successfully receive the downlink packet. To make the decision for having the extra time, the LPHY can estimate how the RX operation is going by querying the status of the preamble detection. If the LoRa modem has detected preamble, then the LPHY should give more time to let the LoRa modem finish the reception. It is noted that this is different with Carrier Activity Detection (CAD), which aims to detect the activity of the radio channel and is independent with the RX procedure.

To realize the above standard scheme, the LPHY needs to have a specific way to recognize whether the LoRa modem detected preamble, and this requires the LoRa modem to provide the information of the status about the preamble detection via a certain register. In SX1272 or SX1276 register set, we can utilize RegModemStat, which provides information about the current modem status. The bit-1 of this register indicates the status of signal synchronization, and is set if the LoRa modem has detected preamble. We therefore generate a new interface to provide this register information to the LPHY. Table 6 shows the modified function set of the hardware-interface entity in the LHAL, which contains a simple change of adding a new function which queries the bit-1 of RegModemStat. Using the new hardware-interface function, we can simply generate a new LHAL command for checking the status of preamble detection.

**Table 6.** The modified function sets of the hardware-interface entity for the RXW prolonging scheme.

| API Functions | Description | Related Registers |
|---|---|---|
| HW_Sleep | Force to stop the current operation | opMode |
| HW_init_radio | initialize the hardware | all |
| HW_Rx | start RX | opMode, rxBuf |
| HW_SetRxConfig | configure RX parameters | freqParms, modParms, enParms, rxBuf |
| HW_GetRxPayload | get RX payload | void |
| HW_Tx | start TX | opMode |
| HW_SetTxConfig | configure TX parameters | freqParms, modParms, enParms |
| HW_SetTxPayload | put TX payload | txBuf, txLen, |
| HW_SetRfTxPower | configure TX power | txPow |
| HW_GetSyncStatus | check if preamble is detected | modemStatus |

Based on the new LHAL command, we modify the LPHY FSM to implement the RX scheme as shown in Figure 7. A new transition from the RX RUN state to the RX RUN state is added. This new transition happens when the LPHY triggers the new LHAL command and recognizes that the LoRa modem has detected preamble at the end of the RXW. In this transition case, the LPHY performs a new action *Extend Timer*, which keeps the RX operation of the LoRa modem by restarting the RX timer. We can see that this RXW prolonging scheme effectively resolves the packet drop issue by adding a single component to each LHAL or LPHY entity. This reveals that our protocol architecture is flexible in adopting new schemes that approach in a cross-layer aspect.

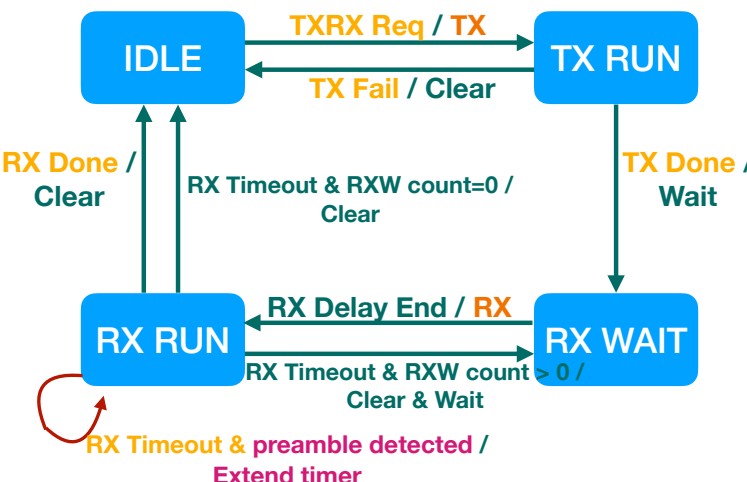

**Figure 7.** The modified LPHY FSM for the RXW prolonging scheme.

## 6. Performance Evaluation

Whereas the proposed protocol has advantageous architecture in a flexible and unified form, it is needed to verify how the proposed protocol performs in practice. The performance of the proposed protocol is measured through test experiments with real-time embedded systems and compared with the conventional software provided by Semtech. Figure 8 shows the overall environment of the test experiments. We used two real-time embedded systems that play roles of a gateway and an end-node, and the LoRaWAN PHY protocol is implemented and runs on the end-node side system. (The source code implemented for the proposed LoRaWAN PHY layer can be accessed in the open source platform [58].) Each embedded system is composed of NUCLEO F446RE as the hardware platform of the PHY protocol software and SX1272 as the LoRa modem. We conduct tests in indoor environments where IoT devices are generally placed.

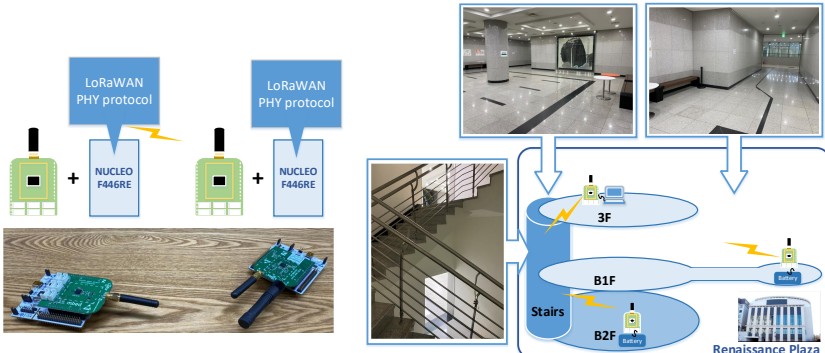

**Figure 8.** Test environments of the protocol evaluation.

The gateway is located at the 3rd floor, and the end-node is located at B1 or B2 floor of the building in Sookmyung Women's University.

Table 7 shows the parameters of the LoRa modulation, which are based on the regional regulation of the LoRaWAN specification in South Korea. The spreading factor is from 7 to 12, which are denoted by DR5-DR0, and the bandwidth and code rate are commonly configured as 125 kHz and 4/5, respectively. The first and second RDs are 1 and 2 s, respectively, and the RXW size is 1 s. To ensure successful acknowledgements at the end-node, the gateway sends a downlink packet 0.1 s after each RXW opens. Table 7 also shows which RXW is used according to the data rate and packet size. RXW1 is mostly used, and RXW2 is used in the case that the transmission time is longer than the RXW1 size.

**Table 7.** LoRa parameter configurations.

| Bandwidth | 125 kHz | | | | | |
|---|---|---|---|---|---|---|
| Code rate | 4/5 | | | | | |
| Default RX Delay | 1 s (RXW1), 2 s (RXW2) | | | | | |
| Length of RXWs | 1 s | | | | | |
| Data rate | DR5 | DR4 | DR3 | DR2 | DR1 | DR0 |
| SF | 7 | 8 | 9 | 10 | 11 | 12 |
| Used RXW (16B) | 1 | 1 | 1 | 1 | 1 | 2 |
| Used RXW (32B) | 1 | 1 | 1 | 1 | 2 | 2 |

The test experiments aim to verify the basic functionality of the class A operations and to estimate Packet Reception Ratio (PRR) and delay time. During the test, the gateway continuously listens for uplink and sends a downlink packet if it receives an uplink packet from the end-node. The PRR is measured from the ratio of the number of transmitted uplink packets to the number of received downlink packets. The PRR for the conventional and proposed protocols is measured in similar RSSI environments. For each test set, the end-node is configure with a parameter set and repeats to send uplink packets for 150 times at the same time zone and location. We accumulate the measured data of test sets to obtain the PRR performance. We also measure round trip time and regard the half as the delay time. Each round trip time is measured with the time interval from the entry of the TX RUN state to the following exit of the RX RUN state.

### 6.1. Regression Test for the Proposed Protocol

We first conducted a regression test for verifying whether the proposed protocol software in the restructured architecture performs in the same level as the conventional protocol software provided by SemTech. Figures 9 and 10 show how the end node receives signal and downlink packets are received with respect to the signal environment. The results reveal that the end-nodes with the conventional and proposed protocols experience the RSSI of similar distributions if the end-nodes are identically located. The results also reveal that the end-node with the proposed protocol successfully receives packets similarly with the conventional protocol.

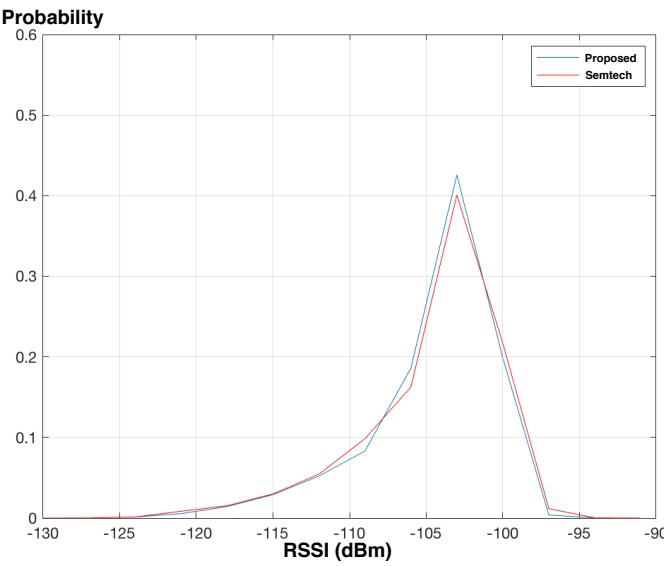

**Figure 9.** The PRR distribution in B1 floor.

In an average perspective, the average PRRs of the proposed protocol are 0.9664 and 0.7873 when the end-node is in B1 floor and B2 floor, respectively, meanwhile the average PRRs of the conventional protocol are 0.9683 and 0.7784. We can see that the end-node in B1 floor mostly receives packets successfully for any SF configurations, since the received signal is relatively strong compared to the one in B2 floor. (The RSSIs in Figure 9 range from −120 to −100 and are on average greater than the ones in Figure 10.) On the other hand, we can observe from Figure 11 that the average PRR is improved as the end-node is configured with higher SF. As we can expect, the PRR depends on how SF is configured, and the end-node or gateway needs to configure SF adaptively when the RSSI is relatively low.

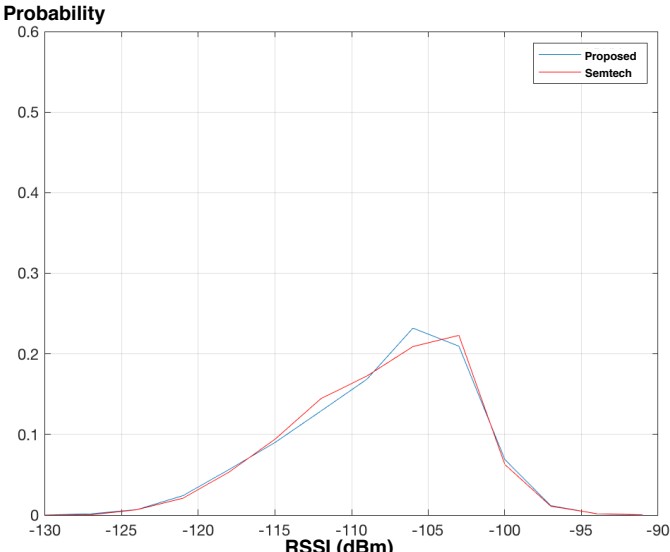

**Figure 10.** The PRR distribution in B2 floor.

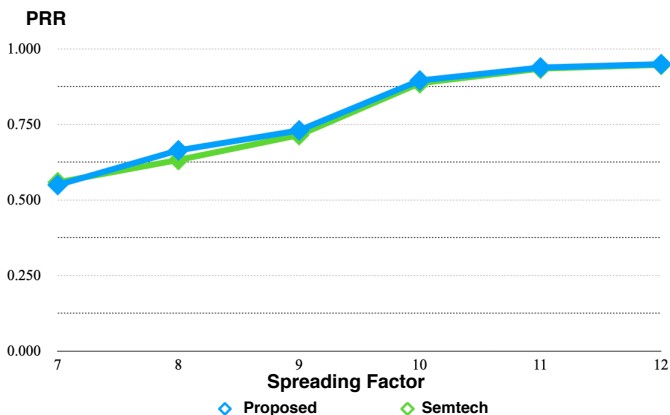

**Figure 11.** PRR vs. SF in B2 floor.

Figure 12, Figure 13 and Table 8 depict delay and transmission time of the conventional protocol and the proposed protocol software when sending 16 and 32 byte packets, respectively. The delay and transmission time generally grow as the end-node configures higher SF due to lower data rate. Compared to the conventional protocol, the proposed protocol consumes more time on transmission, and the delay gets longer by about 30–60 ms. The proposed protocol is designed in layered architecture and multiple layers need to operate for configuring and controlling the LoRa modem, so this makes the whole end-node system consume more CPU resources. (This is trade-off with the advantages of the layered architecture.) However, the increment of the delay and transmission time is negligible since it is less than 5% of the delay or transmission time, so we can consider

that the proposed protocol software also performs in similar level with the conventional protocol in a delay aspect.

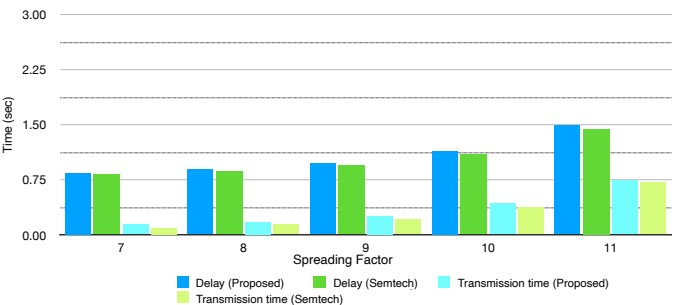

**Figure 12.** Delay and transmission time (16 byte packet).

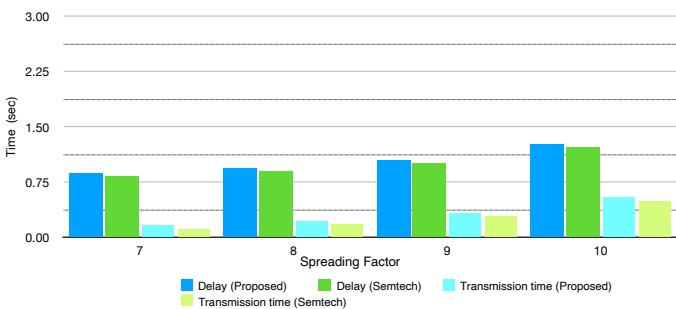

**Figure 13.** Delay and transmission time (32 byte packet).

The regression test proves that the conventional protocol and the proposed protocol software operate with marginal performance differences in various environments. We can therefore consider that the proposed protocol software operates properly as the conventional protocol in terms of configuring and controlling the LoRa modem. Considering that the proposed protocol software has been restructured to have an enhanced architecture, we can conclude that the proposed protocol is advantageous to the conventional protocol when we both consider performance and implementation perspectives.

**Table 8.** Delay and transmission time (s).

|  | **Delay** | | **Transmisson Time** | |
|---|---|---|---|---|
|  | **Semtech** | **Proposed** | **Semtech** | **Proposed** |
| SF7 | 0.82 | 0.85 | 0.10 | 0.14 |
| SF8 | 0.86 | 0.90 | 0.14 | 0.18 |
| SF9 | 0.94 | 0.97 | 0.21 | 0.26 |
| SF10 | 1.10 | 1.14 | 0.38 | 0.42 |
| SF11 | 1.44 | 1.50 | 0.71 | 0.75 |
|  | **Delay** | | **Transmission Time** | |
|  | **Semtech** | **Proposed** | **Semtech** | **Proposed** |
| SF7 | 0.84 | 0.87 | 0.12 | 0.16 |
| SF8 | 0.90 | 0.94 | 0.18 | 0.23 |
| SF9 | 1.02 | 1.05 | 0.29 | 0.33 |
| SF10 | 1.23 | 1.26 | 0.50 | 0.54 |

### 6.2. Performance Improvement by the RXW Prolonging Scheme

To see the impact of the RXW prolonging scheme, we additionally conduct test experiments which can illustrate the substantial improvement of the PRR in RXW1. We focused on the configurations which require long time for transmissions and continue to transmit after the end of the RXW1. The end-node is configured with DR0 and DR1 when sending 16 and 32 byte packets, respectively, and the gateway configures the preamble length as $T_{preamble} = (8 + 4.25)BW/2^{SF}$ and transmits a downlink packet 1.1 s after an uplink packet is received. (So, $T_{preamble}$ is 401 ms for DR0 and 200.5 ms for DR1.) For each test set, the end-node sends 100 packets to the gateway, and is located nearby the gateway so that the RSSI of each received packet becomes above $-50$ dBm.

Figure 14 shows the PRR when the end-node configures RD1 as 100–1300 ms. In the case of DR0, the end-node without the RXW prolonging scheme always fails to receive downlink packets in the RXW1 regardless of RD1 value, since the LPHY aborts the RX operation of the LoRa modem at the end of the RXW1. On the other hands, the end-node with the RXW prolonging scheme successfully receives downlink packets in the RXW1 when the RD1 is from 700 to 1200, since the LPHY recognizes that preamble is detected at the end of the RXW1 and keeps the RX operation. In this case, the end-node fails to receive downlink packets when the RD1 is less or equal to 600 ms because RXW1 ends before the LoRa modem completely detects preamble, and also is larger than 1300 ms because the RXW1 starts after the the gateway begins to transmit the preamble. Similar trend can be observed in case of DR1 and the end-node with the RXW prolonging scheme successfully receives packet in RXW1 when the RD1 is between 400 ms and 1200 ms. The range of the RD1 that the end-node successfully receives varies since the time duration and interval of the preamble transmission get changed.

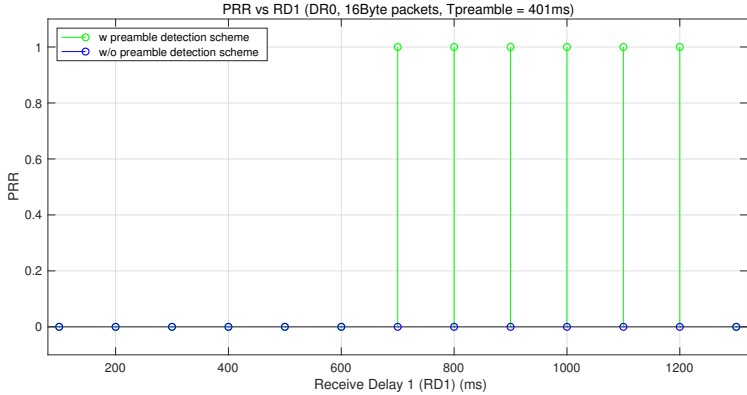

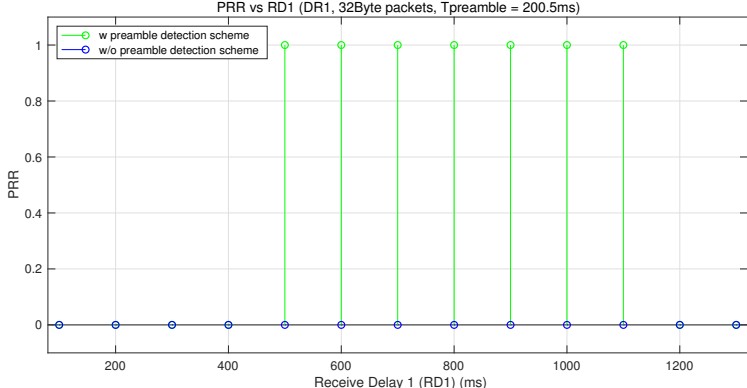

**Figure 14.** PRR when using the RXW prolonging scheme.

As the end-node experiences are enhanced PRR in the RXW1, we can see that the delay times of the downlink packets are shortened in case of the RXW prolonging scheme. The

end-node whose PRR is low in the RXW1 will result in frequent reception of the downlink packet in RXW2, and this causes additional delay time of $RD2 - RD1$. As shown in Table 9, the RXW prolonging scheme makes the delay time to be reduced by 19.7–23%. Accordingly, the experiment results validate that the RXW prolonging scheme, which requires simple modification of our protocol, effectively advances PRR in RXW1 and eventually improves the overall delay time of the downlink packets.

**Table 9.** Delay time (s).

|  | **RXW Prolonging Scheme** | **Conventional RX Scheme** |
|---|---|---|
| DR1, 32 byte | 1.80 | 2.34 |
| DR0, 16 byte | 2.15 | 2.68 |

### 7. Conclusions

For wider application to commercial IoT scenarios, a LoRaWAN solution needs to be flexibly implemented for covering various scenarios and compatible with various models of LoRa modem hardware. We therefore propose a unified architecture of the LoRaWAN PHY protocol, which is flexible in handling service scenarios and inter-working with LoRa modem hardware. The proposed architecture is composed of two sub-layers and each sub-layer is designed based on the concept of FSM for realizing its role efficiently. The LHAL sub-layer is designed to solely have hardware-dependent implementation in the hardware interface entity and abstracted commands in the abstraction entity. The LPHY sub-layer is designed to manage the LoRaWAN PHY procedures triggered by requests from upper layer. The proposed protocol is verified to work properly compared with the conventional protocol software by using real-time operating embedded systems.

To illustrate the flexibility of the proposed architecture, we additionally implement a standard feature that enhances the reception performance based on the proposed protocol architecture. The implementation of this feature requires us to add a simple component in each sub-layer without changing the basic architecture. Through the test experiments with the real-time systems, we show that the implemented scheme can enhance the packet reception ratio and delay time in case of low data rate configurations.

**Author Contributions:** J.P. designed and implemented the architecture and conduct experiments; J.K. analyzed the experimental data and wrote the paper. All authors have read and agreed to the published version of the manuscript.

**Funding:** This research was supported by Institute for Information & communications Technology Promotion (IITP) grant funded by the Korea government(MSIP) (2018-0-00726, Development of Software-Defined Cell/Beam Search Technology for Beyond-5G Systems.) and Sookmyung Women's University Research Grants (1-1903-2003).

**Conflicts of Interest:** The authors declare that there are no conflicts of interest regarding the publication of this paper.

### Abbreviations

The following abbreviations are used in this manuscript:

| | |
|---|---|
| LoRaWAN | Long Range Wide Area Networks |
| IoT | Internet of Things |
| FSM | Finite State Machine |
| LPWAN | Low-Power Wide Area Networking |
| PHY | Physical |
| MAC | Medium Access Control |
| mMTC | massive Machine Type Communication |
| TCP | Transmission Control Protocol |
| NOMA | Non Orthogonal Multiple Access |
| LHAL | LoRaWAN Hardware Abstraction sub-Layer |

| LPHY | LoRaWAN PHY procedure sub-layer |
| TX | Transmission |
| RX | Reception |
| SPI | Serial-to-Peripheral Interface |
| RF | Radio Frequency |
| SF | Spreading Factor |
| RXW | Receive Window |
| RD | Receive Delay |
| SAP | Service Access Point |
| PRR | Packet Reception Ratio |

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
