# Peer review of "An Implementation Design of Unified Protocol Architecture for Physical Layer of LoRaWAN End-Nodes"

_electronics, doi:10.3390/electronics10202550_

Round 1
Reviewer 1 Report
The implemented solution seems solid, by I have concerns about the motivation of the research and its contribution. Please find more detailed comments below.
1. "These proprietary layers garner attention as a prospective wireless communication technology for it can meet the demands on low cost, low battery life and long range." LoRaWAN MAC layer is n
ot proprietary, it is described in an open standard.
2. I do not understand the motivation of the paper. "The PHY layer protocol needs to be designed under a well-organized structure that generalizes various scenarios of the PHY layer procedures s
o that it can operate robustly in any cases." Numerous LoRa devices have been deployed and are successfully operating. Do the authors have any proof that the current implementation of LoRa PH
Y cannot handle the problems that the PHY layer faces? What are the "many scenarios including abnormal events at unexpected moments" that require special attention and additional designs for
LoRaWAN PHY layer?
3. A typo: "The LPHY begins from the IDLE state and and transits" - double "and".
4. In Section "3.3.1 Receiver activity during receive windows" LoRaWAN standard v 1.0.4 states that "If a preamble is detected during one of the receive windows, the radio receiver SHOULD stay active until the downlink frame is demodulated." So the standard currently implies that the devices perform preamble detection and prolonging the receive windows. What is the novelty of soluti
on proposed in Section 5: "A New RX Scheme Based on Preamble Detection" of the paper? It seems that the initial scheme described before Section 5 of the paper contradicts the standard.
Author Response
Dear Reviewer 1,
Thank you for allowing a major revision of our manuscript, with an opportunity to address the reviewers’ comments.
We are uploading our point-by-point response to the comments in the attached document. Please see the attachment.
Best regards,
Jean Park and Juyeop Kim.

Reviewer 2 Report
The authors propose a unified implementation framework for LoRa.
The introduction and abstract should be re-written to explain the objective of the paper clearly. The introduction and abstract sound like the authors are proposing physical layer hardware virtualization. On the contrary, the authors present a unified implementation framework for LoRa.
IBM developed an implementation framework for LoRa called LMIC library. The authors should provide a reasonable explanation of why and how the proposed framework is better than the existing library.
The new reception scheme based on preamble detection seems similar to CAD proposed by LoRaWAN. The authors should either provide a clear description of how it is different from CAD or call it CAD implementation.
The authors should publish their framework on GitHub and provide a citation so that other researchers can use it.
In the introduction, the authors claim that CSS is robust to noise and interference. This claim needs to be supported by references. Furthermore, the term interference is broad, and LoRa's CSS may not be robust (to the best of my knowledge) under some sources of interferences like packet collisions from other LoRa networks or other technologies. The authors should specify the sources of interference CSS is robust against and also provide references so that readers can validate the claim.
The authors can improve the writing in the related work description. Using citations like [6], [7], and [8] as a subject of the sentence does not read well.
Author Response
Dear Reviewer 2,
Thank you for allowing a major revision of our manuscript, with an opportunity to address the reviewers’ comments.
We are uploading our point-by-point response to the comments in the attached document. Please see the attachment.
Best regards,
Jean Park and Juyeop Kim.

Round 2
Reviewer 1 Report
The authors have resolved my comments and have significantly improved the paper. I think it good.
Author Response
Thank you for the positive response. We are appreciated to your valuable comments which surely contributed to improve our paper.
Reviewer 2 Report
The authors should add the reference to the source code in the introduction section, since it is one of the main contributions of the paper.
For related work description in the introduction section, the authors should put the references before the period. For example line 50-52 should be modified as "By adopting the chirp-based spread spectrum modulation, data receptions of the end-nodes and gateways is fundamentally robust to narrowband noise and interference in some degree and can be simply implemented [6, 7]." Currently, the citations seem to be the start of the next sentence.
Author Response
Thank you for allowing a major revision of our manuscript, with an opportunity to address the reviewers’ comments. We have checked each comment and revised the manuscript as following; 1. The authors should add the reference to the source code in the introduction section, since it is one of the main contributions of the paper. => We have added a reference for our source code in the introduction, as following; The source code implemented for the proposed LoRaWAN PHY layer can be accessed in the open source platform. [61] 2. For related work description in the introduction section, the authors should put the references before the period. For example line 50-52 should be modified as "By adopting the chirp-based spread spectrum modulation, data receptions of the end-nodes and gateways is fundamentally robust to narrowband noise and interference in some degree and can be simply implemented [6, 7]." Currently, the citations seem to be the start of the next sentence. => We have missed to check about the position of the reference. We corrected all the references to be shown before the sentence ends. We are appreciated to the detail comments, and we surely think that the comments contribute to improve the quality of the manuscript.